# In Silico Design of a Multiepitope Vaccine Against Intestinal Pathogenic *Escherichia coli* Based on the 2011 German O104:H4 Outbreak Strain Using Reverse Vaccinology and an Immunoinformatic Approach

**DOI:** 10.3390/diseases13080259

**Published:** 2025-08-13

**Authors:** Eman G. Youssef, Khaled Elnesr, Amro Hanora

**Affiliations:** 1The Lundquist Institute at Harbor UCLA Medical Center, Torrance, CA 90502, USA; 2Biotechnology Department, Faculty of Postgraduate Studies for Advanced Sciences, Beni-Suef University, Beni-Suef 62511, Egypt; 3Pathology Department, Faculty of Veterinary Medicine, Beni-Suef University, Beni-Suef 62511, Egypt; khaled.elnesr@vet.bsu.edu.eg; 4Department of Microbiology and Immunology, Faculty of Pharmacy, King Salman International University, Ras Sudr 46611, Egypt; a.hanora@pharm.suez.edu.eg; 5Department of Microbiology and Immunology, Faculty of Pharmacy, Suez Canal University, Ismailia 41511, Egypt

**Keywords:** *Escherichia coli*, multiepitope vaccine, reverse vaccinology, *E. coli* O104:H4, *E. coli* O157:H7, enteroaggregative *E. coli* (EAEC), enterohemorrhagic *E. coli* (EHEC), enteroaggregative hemorrhagic *E. coli* (EAHEC), hemolytic uremic syndrome (HUS), immunoinformatics

## Abstract

Background: While most *Escherichia coli* strains are harmless members of the gastrointestinal microbiota, certain pathogenic variants can cause severe intestinal and extraintestinal diseases. A notable outbreak of *E. coli* O104:H4, involving both enteroaggregative (*EAEC*) and enterohemorrhagic (*EHEC*) strains, occurred in Europe, resulting in symptoms ranging from bloody diarrhea to life-threatening colitis and hemolytic uremic syndrome (HUS). Since treatment options remain limited and have changed little over the past 40 years, there is an urgent need for an effective vaccine. Such a vaccine would offer major public health and economic benefits by preventing severe infections and reducing outbreak-related costs. A multiepitope vaccine approach, enabled by advances in immunoinformatics, offers a promising strategy for targeting HUS-causing *E. coli* (O104:H4 and O157:H7 serotypes) with minimal disruption to normal microbiota. This study aimed to design an immunogenic multiepitope vaccine (MEV) construct using bioinformatics and immunoinformatic tools. Methods and Results: Comparative proteomic analysis identified 672 proteins unique to *E. coli* O104:H4, excluding proteins shared with the nonpathogenic *E. coli* K-12-MG1655 strain and those shorter than 100 amino acids. Subcellular localization (P-SORTb) identified 17 extracellular or outer membrane proteins. Four proteins were selected as vaccine candidates based on transmembrane domains (TMHMM), antigenicity (VaxiJen), and conservation among EHEC strains. Epitope prediction revealed ten B-cell, four cytotoxic T-cell, and three helper T-cell epitopes. Four MEVs with different adjuvants were designed and assessed for solubility, stability, and antigenicity. Structural refinement (GALAXY) and docking studies confirmed strong interaction with Toll-Like Receptor 4 (TLR4). In silico immune simulations (C-ImmSim) indicated robust humoral and cellular immune responses. In Conclusions, the proposed MEV construct demonstrated promising immunogenicity and warrants further validation in experimental models.

## 1. Introduction

*Escherichia coli*, a gram-negative bacterium, is a common constituent of the normal microbiota in humans and animals. It primarily colonizes the gastrointestinal tract, particularly the large intestine, soon after birth. While typically part of the normal gut flora, *E. coli* can also contribute to various illnesses, both intestinal and extraintestinal [1]. The Centers for Disease Control and Prevention (CDC) categorize intestinal *E. coli* causing gastroenteritis into various pathotypes, including enteroaggregative *E. coli* (EAEC), enteroinvasive *E. coli* (EIEC), enterotoxigenic *E. coli* (ETEC), enteropathogenic *E. coli* (EPEC), and enterohemorrhagic *E. coli* (EHEC). It is important to note that enterohemorrhagic *E. coli* (EHEC) is a subset of Shiga toxin–producing *E. coli* (STEC), although these terms are sometimes used interchangeably [2]. Each pathotype has a unique pathogenesis and is characterized by specific O:H serotypes, contributing to various epidemiological patterns and associated pathological conditions. Nevertheless, their interchangeable nature complicates the differentiation of traits within each subclass [3].

While some pathogenic strains of *E. coli* had been recognized earlier, *E. coli* O157:H7 was first identified as a major cause of foodborne illness in 1982, following outbreaks of bloody diarrhea in the U.S. This strain, a type of enterohemorrhagic *E. coli* (EHEC), has since become a significant public health concern [4,5,6,7]. Subsequently, sporadic cases emerged, showing severe colonic and/or renal diseases, such as hemolytic uremic syndrome (HUS) [6,8]. EHEC poses a significant global public health concern, primarily as a foodborne illness. It spreads through contaminated food during meat processing (slaughter process) or via EHEC-contaminated water reaching agricultural produce. The consumption of raw or undercooked beef, especially ground beef (hamburger), serves as a common transmission route for EHEC O157:H7. Outbreaks have also been linked to contaminated foods such as radish sprouts (e.g., the Sakai city incident in Japan in 1996), lettuce, spinach, strawberries, and tainted water [9,10,11,12,13,14,15]. Human fecal contamination of food and seeds may additionally contribute, particularly in developing countries [16].

While EHEC O157:H7 remains the most common and prevalent serotype associated with sporadic HUS cases [17], other non-O157:H7 serotypes are also noteworthy. In May 2011, a rare and novel O104:H4 serotype outbreak captured global attention for its significant impact, notably in Germany [18,19], France [20], and other European countries. Although there were only a few reported cases in Canada and the United States, these were primarily individuals who had recently visited Europe before becoming ill. The 2011 outbreak of *E. coli* O104:H4 in Europe displayed unusual virulence and lethality patterns [21]. Although a related *E. coli* O104:H4 strain was reported in a 2009 outbreak in the Republic of Georgia, that strain exhibited a different molecular profile, lacked some key virulence factors (such as the Shiga toxin gene), and showed lower levels of antibiotic resistance. This indicates that while both strains share the O104 serogroup, they represent distinct lineages with different pathogenic potentials.

Transmission during the 2011 outbreak was primarily through consumption of fenugreek sprouts contaminated with *E. coli* O104:H4 [22,23,24,25], and the outbreak showed limited zoonotic potential [26,27,28,29]. HUS cases began to cluster in northern Germany in May 2011, peaking and declining by July due to control measures. The Robert Koch Institute (RKI) reported 3842 cases, including 855 cases of HUS and 53 fatalities [19]. Approximately one month later, a smaller outbreak involving the same *E. coli* O104:H4 strain occurred in France [17,20].

This 2011 STEC outbreak, as reported by the WHO, affected 4075 people in 16 countries, resulting in 908 cases of HUS and 50 deaths, with a mortality rate of 1.23%. The mortality rate for HUS caused by *E. coli* O104:H4 was notably greater at 3.74% [30]. Approximately 90% of HUS cases occurred in adults, with about two-thirds of those in females. Around 10% of HUS cases were reported in children [19]. Transmission likely occurred through contact with infected individuals. This strain is more likely to cause severe disease in adults than in children [31].

The O104:H4 strain is a rare hybrid of enteroaggregative and Shiga toxin-producing *E. coli* resulting from genetic integration of virulence factors from both types, termed enteroaggregative hemorrhagic *E. coli* (EAHEC) [32]. This strain, identified in the 2011 outbreak, belonged to an EAEC lineage that acquired genes for Stx2 and antibiotic resistance [32,33,34,35,36]. It exhibits aggregative adherence fimbriae (AAFs) similar to those of EAEC, facilitating strong adhesion to food surfaces and the intestinal wall and enhancing its persistence and pathogenicity. This enhanced adhesion may also increase the absorption of Shiga toxin (Stx), the key factor contributing to the destruction of gut epithelial cells, leading to more severe symptoms, including abdominal cramps, bloody diarrhea, and HUS [2,37,38]. The strain produces extended-spectrum beta-lactamases (ESBL) [39,40], contributing to its ability to colonize and release toxins in the gut [34]. Infections typically last approximately two weeks, with an increased risk of developing HUS and mortality if treatment is inadequate [19,41,42].

Treatment for infections caused by EHEC, particularly strains such as O104:H4 and O157:H7, primarily involves supportive care due to the lack of specific antimicrobial therapies, which could worsen symptoms by triggering toxin release [43]. Supportive measures include hydration and electrolyte balance management, often requiring hospitalization for close monitoring. In severe cases, such as those leading to HUS, interventions such as blood transfusions and renal replacement therapy may be necessary [44,45,46,47]. The treatment of HUS caused by these infections is particularly challenging and has unpredictable outcomes. Traditional antibiotics do not effectively treat the disease and may even exacerbate symptoms by increasing Shiga toxin release. The complex pathogenesis of EHEC, involving adherence to intestinal cells and toxin production, complicates therapeutic strategies. Research into antibody-based therapies targeting Shiga toxins has shown promise in reducing toxin-mediated damage, although their clinical effectiveness remains uncertain and requires further study [43,48,49,50,51].

Given the lack of effective treatments and the serious clinical consequences associated with EHEC serotypes O104:H4 and O157:H7, the CDC categorizes them as high-risk pathogens capable of causing outbreaks with significant illness and complications [2,52]. Consequently, researchers are increasingly focusing on vaccine development as a preventive strategy, especially for travelers and high-risk populations [53]. Vaccines targeting specific EHEC serotypes, such as O157:H7 and O104:H4, aim to prevent infection and reduce symptom severity. The cost-effectiveness of vaccination in mitigating healthcare burdens from EHEC outbreaks and complications like HUS highlights their public health importance [7,54].

Despite ongoing efforts, no approved vaccines currently exist for EHEC infections. Novel research is crucial to identify effective vaccine candidates that specifically target pathogenic strains like O157:H7 and O104:H4 while preserving the normal gut flora. However, the genetic and antigenic diversity of EHEC strains remains a major challenge for vaccine development. Although infrequent, the severity and unique virulence profile of O104:H4 justify its inclusion in vaccine research efforts, particularly as a model for hybrid EHEC/EAEC strains. Developing multivalent or cross-protective vaccines that account for this diversity is a critical public health priority to prevent infections and reduce severe complications like HUS [43].

Reverse vaccinology, which leverages genomic data and bioinformatics to identify potential vaccine targets, has accelerated vaccine candidate discovery. In particular, multiepitope vaccines (MEVs) are gaining interest due to advances in computational methods for predicting epitopes and immune responses [55,56]. MEVs combine selected epitopes from multiple proteins into a single construct to enhance immunogenicity and efficacy. This approach has been widely studied across various pathogens and holds promise for next-generation vaccine development [57,58,59,60,61,62,63,64,65,66].

### Aim of Work

To design a multiepitope vaccine based on proteins uniquely present in the *E. coli* O104:H4 proteome that are conserved in the predominant EHEC strain *E. coli* O157:H7, with the goal of targeting pathogenic strains specifically while preserving the commensal gut microbiota.

## 2. Materials and Methods

### 2.1. Data Retrieval and Comparative Proteomic Analysis Using a Reverse Vaccinology Approach

To identify potential vaccine targets specific to the pathogenic *Escherichia coli* O104:H4 strain, a reverse vaccinology approach was employed. Complete proteomes of *E. coli* O104:H4 strain 2011C-3493 and the commensal *E. coli* K-12 substr. MG1655 were retrieved from the NCBI FTP site using GenBank assembly accession numbers GCF_000299455.1 (CP003289.1) and GCA_000005845.2 (U00096.3), respectively.

A comparative proteomic analysis using standalone BLASTP v2.2.26 was conducted to identify proteins unique to the pathogenic O104:H4 strain. Proteins showing >80% coverage and >40% similarity to those in the K-12 strain were excluded, along with proteins shorter than 100 amino acids. This approach aimed at eliminating proteins shared with commensal *E. coli*, thereby focusing on strain-specific proteins that may contribute to pathogenicity while minimizing disruption to the normal gut flora. Proteins shorter than 100 amino acids were excluded because they are more likely to be non-functional, hypothetical, or result in spurious alignments, which could obscure the identification of meaningful virulence factors.

### 2.2. Bioinformatic Characterization of the Candidate Proteins

The unique proteins identified were subjected to a series of bioinformatic analyses to evaluate their suitability as vaccine targets. Since ideal vaccine candidates are typically extracellular or outer membrane proteins, subcellular localization was first assessed using the **P-SORTb tool** (https://www.psort.org/psortb/ (accessed on 1 June 2023)) [67]. Seventeen proteins were predicted to be located in the outer membrane and extracellular space. Subsequently, these proteins were further evaluated based on several key criteria, including antigenicity using **VaxiJen** (https://www.ddg-pharmfac.net/vaxijen/VaxiJen/VaxiJen.html (accessed on 1 June 2023)) [68], conservation among other *E. coli* O104:H4 strains and related intestinal pathogenic strains (e.g., O157:H7) to ensure broad coverage, similarity to human proteins to minimize cross-reactivity, and functional relevance related to virulence or pathogenicity. Additionally, transmembrane prediction was conducted using **TMHMM 2** (https://services.healthtech.dtu.dk/services/TMHMM-2.0/ (accessed on 1 June 2023)) [69] to ensure accessibility of epitopes.

After completing the reverse vaccinology-driven selection process, the most promising vaccine candidates were selected for epitope mapping. Prior to epitope prediction, signal peptide regions were identified using **SignalP** (https://services.healthtech.dtu.dk/services/SignalP-5.0/ (accessed on 1 June 2023)) [70] and **LipoP** (https://services.healthtech.dtu.dk/services/LipoP-1.0/ (accessed on 1 June 2023)) [71] and subsequently excluded to ensure accurate epitope identification within the mature protein regions.

### 2.3. Epitope Mapping

Epitope mapping was performed to identify specific immunogenic regions within the selected proteins that can effectively elicit a targeted immune response without causing allergenic or toxic effects.

#### 2.3.1. Linear B-Lymphocyte (LBL) Epitope Prediction

Linear B-cell epitopes for each target protein were predicted using multiple computational tools to enhance prediction reliability:

**ABCpred** (https://webs.iiitd.edu.in/raghava/abcpred/ABC_submission.html (accessed on 1 July 2023)) [72,73] with a cutoff score ≥ 0.8 at a length of 16 amino acids.

**BepiPred-2.0** was obtained from the Immune Epitope Database (IEDB) (http://tools.iedb.org/bcell/ (accessed on 1 July 2023)) [74].

**BCEPRED** (https://webs.iiitd.edu.in/raghava/bcepred/bcepred_submission.html (accessed on 1 July 2023)) [75] at a threshold of −3 to 3 are based on both accessibility and surface exposure.

The predicted LBL epitopes were further screened:Isotype prediction was performed using **IgPred** (https://webs.iiitd.edu.in/raghava/igpred/pep-fix-pred.html (accessed on 1 August 2023)), which predicts specific B-cell isotypes (IgG, IgA, or IgE) using a fixed-length epitope model with a threshold of 0.7 [76].Allergenicity was evaluated using **Allertop v.2** (https://www.ddg-pharmfac.net/allertop/ (accessed on 1 August 2023)) [77].Toxicity was assessed via **ToxinPred** (http://crdd.osdd.net/raghava/toxinpred/ (accessed on 1 August 2023)) [78].Virulence potential was analyzed using **VirulentPred** (https://bioinfo.icgeb.res.in/virulent/submit.html (accessed on 1 August 2023)).

The selection of the final LBL epitopes was based on the frequency of predication across different software tools, isotype predication (IgG or IgA), absence within signal peptides, and their non-allergenic, non-toxic, and virulent characteristics.

#### 2.3.2. Cytotoxic T-Lymphocyte (CTL) Epitope Prediction

**NetMHC-4.0** (https://services.healthtech.dtu.dk/services/NetMHC-4.0/ (accessed on 1 July 2023)) was used to predict cytotoxic T-cell epitopes [79,80]. Predictions of MHC-I binding were performed for 23 HLA-A, HLA-B, HLA-C, and HLA-E alleles with a length of 9 mer. Epitopes were selected based on a threshold for strong binding of 2% (adjusted rank ≤ 2); weak binders (adjusted rank ≤ 5%) were excluded.

The immunogenicity of the predicted CTL epitopes was assessed using **Class I immunoreactivity** from the IEDB analysis resource (http://tools.iedb.org/immunogenicity/ (accessed on 1 July 2023)) [81].

To evaluate TAP transport and proteasomal cleavage, **the NetCTL1.2** server (http://www.cbs.dtu.dk/services/NetCTL (accessed on 1 July 2023)) was used [82]. Epitopes with an epitope identification score > 0.75 were considered, indicating high-quality proteasomal cleavage and efficient TAP transport, which are crucial for antigen presentation to CTLs.

The selection of the CTL epitopes was based on predicted binding to multiple alleles of MHC-I, immunogenicity, favorable TAP cleavage IC50 values, strong antigenicity scores, and predictions indicating non-allergenicity, non-toxicity, and virulence potential.

#### 2.3.3. Helper T-Lymphocyte (HTL) Epitope Prediction

**NETMHCII_pan 4.0** (https://services.healthtech.dtu.dk/services/NetMHCIIpan-4.0/ (accessed on 1 July 2023)) was used to predict possible helper T-cell epitopes in each of the targeted proteins [83]. The prediction of MHC-II binding was performed for 20 HLA-DR alleles with a length of 15 mer. Epitopes were selected based on a threshold for strong binding of 2% (adjusted rank ≤ 2); weak binders (adjusted rank ≤ 5%) were excluded.

For each predicted HTL epitope, the ability to induce cytokine production was assessed:The **IFNepitope** server (http://crdd.osdd.net/raghava/ifnepitope/predict.php (accessed on 1 August 2023)) was used to predict the ability of HTL epitopes to induce interferon-gamma (IFN-γ) production [84].The **IL4pred** server (https://webs.iiitd.edu.in/raghava/il4pred/design.php (accessed on 1 August 2023)) with a threshold of 0.2 was used to assess HTL epitopes for interleukin-4 (IL-4) production [85].The **IL10pred** server (https://webs.iiitd.edu.in/raghava/il10pred/predict3.php (accessed on 1 August 2023)) was utilized to predict IL-10 production by HTL epitopes [86].

The selection of HTL epitopes was based on predicted binding to multiple MHC-II alleles, their ability to induce key cytokines (IFN-γ, IL-4, and IL-10), high antigenicity scores, non-allergenic and non-toxic profiles, and demonstrated virulence potential.

### 2.4. Construction of a Multiepitope Vaccine (MEV)

A potential MEV was constructed by linking selected LBL, CTL, and HTL epitopes identified during epitope mapping. These epitopes were connected using KK, AAY, and GPGPG amino acid linkers, respectively, to ensure effective in vivo separation [58,87]. To maximize immunogenicity, the assembled epitopes were conjugated with various adjuvants known to enhance different aspects of the immune response (to be further discussed in Section 3), including a partial sequence of human β-defensin (GIINTLQKYYCRVRGGRCAVLSCLPKEEQIGKCSTRGRKCCRRKK), cholera toxin subunit B (CTXB) partial (AAISMAN), *S. dublin* flagellin, and RS09 (APPHALS).

Additionally, the Pan DR epitope (PADRE) sequence (AKFVAAWTLKAAA) was strategically incorporated into the construct to augment immune responses and as a stabilizer [88]. The adjuvants were linked with epitope assembly via an EAAAK linker, optimizing the functionality of the MEV construct [89,90].

### 2.5. Physicochemical Properties, Solubility Profile, Antigenicity, and Allergenicity of the Constructed MEV

The physicochemical properties were evaluated using the **ExPASy ProtParam** tool (https://web.expasy.org/protparam/ (accessed on 1 September 2023)) [91]. This tool calculates various properties, including molecular weight, theoretical isoelectric point (pI), estimated half-life, instability index, aliphatic index, and grand average hydropathicity (GRAVY). The solubility profile was predicted using two different tools: **SOLpro** (https://scratch.proteomics.ics.uci.edu/ (accessed on 1 September 2023)) [92] and **Protein-Sol** (https://protein-sol.manchester.ac.uk/ (accessed on 1 September 2023)) [93]. A construct was considered highly soluble if its solubility score was greater than 0.5. Additionally, the presence of signal peptides and transmembrane regions was assessed, along with the construct’s similarity to human proteins to minimize potential autoimmune responses.

### 2.6. Secondary Structure Prediction

The secondary structures of the MEV constructs were predicted using **PSIPRED** (http://bioinf.cs.ucl.ac.uk/psipred/ (accessed on 1 September 2023)) [94], a PSI-blast-based secondary structure prediction tool.

### 2.7. Tertiary Structure Prediction, Refining and Validation

The tertiary (3D) structure was modeled using the de novo modeling tool **I-TASSER** (https://zhanggroup.org/I-TASSER/ (accessed on 1 October 2023)) [95]. Visualization of the 3D structure was performed using **PyMOL** education (https://pymol.org/edu/ (accessed on 1 October 2023)) [96]. The modeled 3D structure was subjected to a refinement process using the GalaxyRefine tool on the **GlaxyWEB** server (https://galaxy.seoklab.org/cgi-bin/submit.cgi?type=REFINE (accessed on 1 October 2023)). The best module was selected based on parameters including GDT-HA, RMSD, MolProbity, Clash score, Poor rotamers, and Rama favored [97]. Stereochemical quality was assessed based on dihedral angles (ψ and φ) using a Ramachandran plot provided by the **PROCHECK** application (https://saves.mbi.ucla.edu/) [98]. The refined model was validated by calculating the z score using the **ProSA** server (https://prosa.services.came.sbg.ac.at/prosa.php (accessed on 1 October 2023)) to ensure that it fell within the typical range for native proteins of similar size [99]. To confirm the validated model, **ERRAT** (https://saves.mbi.ucla.edu/ (accessed on 1 October 2023)) and **Verify3D** (https://saves.mbi.ucla.edu/ (accessed on 1 October 2023)) were utilized [100,101].

### 2.8. Disulfide Engineering

Disulfide engineering was assessed using the **Disulfide by Design 2** online server (http://cptweb.cpt.wayne.edu/DbD2/index.php (accessed on 1 October 2023)), employing default settings [102].

### 2.9. Prediction of Glycosylation

Glycosylation was predicted using the **GlycoPP v1.0** online server (https://webs.iiitd.edu.in/raghava/glycopp/submit.html (accessed on 1 October 2023)). Both N-linked and O-linked glycosylation predictions were conducted using prediction based on Binary Profile of Patterns (BPP) with the default parameters [103].

### 2.10. Molecular Docking of MEV

The finalized MEV construct was molecularly docked with Toll-like Receptor 4 (TLR4) (PDB ID: 4G8A) using the **ClustPro** server (https://cluspro.bu.edu/home.php (accessed on 1 October 2023)). The PDB files were submitted to the server using the default settings [104,105]. The best TLR4 vaccine-docked complex with the lowest energy was selected, and the resulting structure was visualized using the PyMOL tool.

### 2.11. Immune Stimulation

To simulate the real-life immune response, computational immune stimulation was conducted using the **C-ImmSim** online tool (https://kraken.iac.rm.cnr.it/C-IMMSIM/index.php?page=1 (accessed on 1 October 2023)). The default parameters were used for simulation, except for the time steps, which were set at 1, 84, and 170 (time step 1 represents the initial injection at time = 0, with subsequent injections every 8 h). The simulation was conducted with three injections, each four weeks apart, corresponding to the recommended interval between doses for most commercial vaccines [106].

## 3. Results

The initial step in designing a vaccine against pathogenic *E. coli* was identifying target proteins using a reverse vaccinology approach, as illustrated in Figure 1.

### 3.1. The Identification of Potential Vaccine Candidates That Are Not Shared with Nonpathogenic E. coli, Have Outer Membrane, Are Antigenic and Are Important for Virulence

To avoid disrupting the normal gut microbiota—which are essential for maintaining gut health—we screened the entire proteome of *E. coli* O104:H4 using reciprocal BLAST analysis. Proteins shared with the nonpathogenic *E. coli* strain K-12 MG1655 were excluded, as detailed in Section 2. This exclusion process resulted in a refined dataset of 672 proteins (Appendix A).

Next, subcellular localization was predicted using P-SORTb to identify proteins likely to be exposed on the bacterial surface. This step revealed 17 proteins localized to the outer membrane or extracellular space (Table 1), making them initial candidates for vaccine development.

To further narrow down these 17 proteins to the most promising vaccine candidates, we applied a set of stringent selection criteria:(1)Antigenicity—Proteins with a VaxiJen score > 0.4 were considered antigenic.(2)Transmembrane Helix—Proteins without predicted transmembrane helices were prioritized, ensuring full surface accessibility.(3)Conservation—Candidates conserved across multiple *E. coli* O104:H4 strains and also present in other pathogenic strains, such as *E. coli* O157:H7, were selected to ensure broad-spectrum protection.(4)Specificity—Proteins not shared with other nonpathogenic *E. coli* strains, such as HS and W3110, were chosen to confirm their pathogenic-specific nature and minimize cross-reactivity with commensal strains.(5)Human Proteome Exclusion—Proteins were checked against the human proteome via the NCBI database to avoid potential autoimmunity.

Following this rigorous filtration process, four proteins emerged as strong vaccine candidates due to their virulence relevance and favorable immunogenic profiles: copper resistance protein B (CopB), long polar fimbrial protein (LpfD), putative outer membrane protein Lom (LomP), and hypothetical protein O3K_20405 (Hcp_VI), as highlighted in Table 1.

### 3.2. Identification and Mapping of B-Cell and T-Cell Epitopes

Following the screening and selection of potential protein candidates from the *E. coli* O104:H4 proteome, the subsequent step in vaccine design involved scrutinizing each protein for potential linear B-cell, T-helper, and T-cytotoxic lymphocyte epitopes, separately.

#### 3.2.1. Predication of Linear B-Cell Epitopes

The linear B-cell epitopes, which are crucial for inducing a humoral immune response, were predicted using three distinct online servers: ABCpred, BepiPred-2.0, and BCEPRED. These servers identified 39, 38, 36, and 23 epitopes for copB, LpfD, LomP, and Hcp_VI, respectively, as detailed in Appendix A. Predicted epitopes were then evaluated for signal peptides, antigenicity, allergenicity, toxicity, virulence potential, and immunoglobulin isotype. Optimal epitopes were selected based on (1) consistency across multiple prediction tools to ensure accuracy, (2) high antigenicity score (VaxiJen score > 0.6), (3) nonallergenic and non-toxic characteristics, (4) virulence attributes, (5) absence of a signal peptide, and (6) ability to induce IgG or IgA isotypes. Ten LBL epitope sequences were finalized from all four proteins, and their sequences and properties are detailed in Table 2.

#### 3.2.2. Predication of Cytotoxic T-Lymphocyte Epitope

The prediction of CTL epitopes was performed using NetMHC-4.0, where different alleles, including HLA-A, HLA-B, HLA-C, and HLA-E, were screened for MHC-I binding epitopes. Specifically, 108, 118, 83, and 53 epitopes were predicted for copB, LpfD, LomP, and Hcp_VI, respectively, as detailed in Appendix A. The selection of CTL epitopes was based on consistent prediction across multiple alleles, immunogenicity, and favorable TAP cleavage IC_50_ values. One CTL epitope was selected from each protein, totaling four CTL epitopes, as listed in Table 3.

#### 3.2.3. Predication of Helper T-Lymphocyte Epitope

HTL epitopes were predicted using the NETMHCII_pan 4.0 server across various HLA-DR alleles, yielding 88 epitopes (21 for copB, 30 for LpfD, 24 for LomP, and 13 for Hcp_VI) as detailed in Appendix A. From these, three HTL epitopes were selected (Table 4) based on their conservation across alleles, and ability to induce IL-4 and/or IL-10 cytokines, which are critical in bacterial infections.

Overall, the selected CTL and HTL epitopes demonstrate high antigenicity (VaxiJen score > 0.6), and are nonallergenic, nontoxic, and virulent.

### 3.3. Design of the MEV Construct

All 17 epitopes were interconnected using appropriate linkers. LBL epitopes were joined together using a bilysine (KK) linker. KK linkers are specifically recognized by cathepsin B, a lysosomal protease involved in processing antigenic peptides for their presentation on the cell surface via MHC-II-restricted antigen presentation [107,108]. For the CTL and HTL epitopes, alanine-tyrosine (AAY) and glycine-proline-glycine-proline (GPGPG) linkers were utilized, respectively. The AAY linker acts as a cleavage site for proteasomes in mammalian cells, facilitating the efficient separation of epitopes within cells [64,109,110]. GPGPG linkers are known to stimulate HTL responses. These linkers play a critical role in enhancing the immunogenicity of MEVs and are essential tools for overcoming junctional immunogenicity, thereby restoring the immunogenic potential of individual epitopes [60,62,111,112,113,114,115,116].

Four distinct MEV constructs were designed to elicit diverse immune responses, each incorporating different adjuvants. The selected adjuvants included partial human β-defensin (UniProt ID: P81534), which acts as an agonist for TLR1, 2, and4 [63,117]; cholera toxin B subunit (CTXB), a T-helper type 1 (Th1) agonist with anti-inflammatory properties [60]; *Salmonella dublin* flagellin, a TLR5 agonist known to stimulate IFN-γ and TNF-α production [118]; and RS09, a synthetic peptide serving as a TLR4 agonist [119]. Additionally, PADRE, a universal peptide, was included as a T-helper epitope separated by the EAAK sequence, which is known to enhance the functionality of MEV constructs when conjugated with other adjuvants [59,89,120]. Figure 2 illustrates the finalized MEV constructs, named *Ecoepvc*, following the integration of the various adjuvants.

### 3.4. Features of the Construct

The physicochemical properties of each construct were evaluated and are summarized in Table 5. All MEV constructs exhibited high stability (instability index < 40), demonstrated excellent thermal stability, as assessed by the aliphatic index, and displayed hydrophilic properties, as indicated by the GRAVY. Predictions from the Protein-Sol and SOLpro servers indicated high solubility of all vaccine constructs. Furthermore, the constructs displayed high antigenicity scores, confirming their non-allergenic and non-toxic nature. The predicted secondary structures of all constructs are shown in Figure 3.

### 3.5. Tertiary Structure Modeling and Structure Refining and Validation

3D modeling was conducted using the i-Tasser online server, and the model with the highest C-score was selected. Subsequently, the modeled structures were refined using GalaxyWEB software. This process involved removing steric clashes, reconstructing side chains, and generating several models. The refinement process considered parameters such as global distance test-high accuracy (GDT-HA), root-mean-square deviation (RMSD), MolProbity, and the Ramachandran favored score. Models with lower RMSD values (between 0 and 1.2 Å), MolProbity, Clash score, and Poor rotamer values and higher GDT-HA and Rama favored values were prioritized for their improved stability and of high quality for further validation (Figure 4A).

Validation of the refined vaccine constructs was performed using a Ramachandran plot, which illustrates the percentage of amino acid residues within favored, generously allowed, additionally allowed, and disallowed regions. Post-refinement, all constructs showed a significant improvement in the percentage of residues within favored regions. *Ecoepvc3* demonstrated the highest quality, with 88% of residues located in the most favored regions (Figure 4B), followed by *Ecoepvc4* with 86.6%, *Ecoepvc1* with 84.5%, and *Ecoepvc2* with 82.1% (Appendix A). A high-quality protein model typically exhibits approximately 90% of residues in favored regions; therefore, *Ecoepvc3*, being closest to this benchmark, was selected for further analysis [121].

To further assess the structural integrity of *Ecoepvc3*, its 3D atomic model was evaluated for compatibility with its amino acid sequence using the Verify 3D and ERRAT tools. The Verify 3D tool assesses the compatibility between the 3D model and its linear sequence, while ERRAT identifies regions of potential error based on non-random atomic interactions [100,122,123]. Results from both validation tools confirmed the high quality of the *Ecoepvc3* model; see Figure 5.

### 3.6. Disulfide Bond Engineering Results

*Ecoepvc3* presents several potential disulfide bonds, with nine bonds identified within the favored energy range of <2 and chi3 angles ranging from −87 to + 97 ±30 degrees (Figure 6). These bonds involve pairs: SER11-ALA522, LEU18-GLY515, LEU25-ALA508, LEU109-ASN177, KYS136-SER139, ASN285-ASN290, KYS315-GLU320, SER324-ALA341, and ALA347-LEU353.

Predicting and understanding disulfide bonds in *Ecoepvc3* is critical because these bonds stabilize protein structures, ensuring proper folding and resistance to degradation. This structural insight is crucial for optimizing vaccine stability and efficacy, which are essential for biotechnological and therapeutic applications [57].

### 3.7. Glycosylation Site Prediction

Predicting glycosylation sites is important in MEV design because it impacts antigen immunogenicity, stability, and immunomodulatory properties. Identifying potential glycosylation sites ensures optimal antigen presentation and immune recognition, thereby enhancing vaccine efficacy. Avoiding the glycosylation of critical epitopes prevents interference with antigenic presentation and reduces the risk of eliciting undesired immune responses [124]. GlycoPP v1.0 analysis revealed multiple N-linked glycosylation sites primarily located in the adjuvant flagellin head and tail regions, as indicated in Table 6, rather than concentrated in the epitope region. However, O-linked glycosylation sites were predominantly located, with 11 sites predicted within the epitope region, as shown in Table 7. This localization is advantageous because it minimizes potential glycosylation interference with epitope functionality, preserving their immunogenicity and enhancing vaccine specificity.

### 3.8. Molecular Docking of MEV with TLR4

The refined structure of *Ecoepvc3* was docked against Toll-like Receptor 4 (TLR4), which is essential for understanding how vaccine components interact with this receptor at the molecular level. As shown in Figure 7, *Ecopev3* strongly binds to the epitope regions of TLR4.

### 3.9. Predicted Immune Responses Induced by the MEV 

*Ecopev3* demonstrated an enhanced immune response, particularly after boosting, characterized by elevated concentrations of immunoglobulins IgM and IgG1, with IgG2 levels remaining unaffected (Figure 8). The vaccine also notably induced a substantial increase in T-helper cells, predominantly T-helper 1 (Th1) rather than T-helper 2 (Th2) cells. Despite incorporating PADRE sequences, which are recognized for enhancing Th2 responses, into the vaccine construct at multiple sites, the C-IMMSUM analysis shown in Figure 8 revealed its ineffectiveness in promoting Th2 differentiation.

## 4. Discussion

The primary objective of this study was to identify strategic protein candidates for the development of an *E. coli* vaccine, focusing on proteins exclusive to pathogenic strains, conserved among HUS-causing variants, and essential for their pathogenicity. Given the challenging treatment landscape presented by the highly adhesive *E. coli* O104:H4 outbreak strain, our research emphasizes proteins critical for adhesion, attachment, and colonization promotion while also evaluating proteins that may disrupt cellular integrity and homeostasis. Additionally, we prioritized candidates conserved across various strains, including other EHEC strains, particularly the O157:H7 serotype. Careful selection criteria ensured that these candidates were less conserved or absent in three different commensal *E. coli* strains.

Our investigation identified four pivotal protein candidates—copper resistance protein B (CopB), long polar fimbrial protein (LpfD), putative outer membrane protein Lom (LomP), and hypothetical protein O3K_20405 (Hcp_VI)—all of which play significant roles in the infection process.

Copper resistance protein B (CopB) in *E. coli* is essential for bacterial adaptation and survival in environments with elevated copper levels, including host tissues during infection. CopB’s role is copper detoxification by facilitating the efflux or sequestration of copper ions, thereby maintaining cellular homeostasis and protecting against copper-induced toxicity. The function of CopB extends to contributing to virulence mechanisms, potentially aiding in the bacterium’s ability to evade host immune responses and persist within the host [125,126,127,128,129]. Targeting CopB in vaccine development could disrupt these adaptive strategies, potentially reducing *E. coli* virulence and enhancing therapeutic strategies against infections involving copper-rich environments.

Long polar fimbrial protein (LpfD) plays a central role in EHEC pathogenesis by mediating adhesion to intestinal epithelial cells. This protein facilitates the formation of microcolonies and biofilms on the mucosal surface, promoting persistent colonization. Its adhesive function is particularly relevant in O104:H4 and other EHEC strains, such as O157:H7, O26, O111, and O145, where prolonged intestinal attachment correlates with extended Shiga toxin (Stx) shedding, exacerbating disease severity and duration [130,131]. Therefore, targeting LpfD may help prevent early colonization and reduce the clinical impact of EHEC infections [132,133,134]. The high conservation of LpfD across O104:H4 strains, as shown in Table 1, supports its utility as a universal vaccine antigen.

The putative outer membrane protein Lom (LomP) is notably more prevalent in O157:H7 strains than in O104:H4 strains, as highlighted in Table 1. It plays a vital role in maintaining outer membrane integrity and contributes to immune evasion. In pathogenic EHEC, including O104:H4, LomP supports adhesion to intestinal epithelial cells, facilitating microcolony formation and enhancing virulence [135,136]. Given its higher prevalence in O157:H7 strains—commonly associated with severe HUS—LomP is a valuable vaccine target. Its inclusion in multivalent formulations could provide cross-protection against HUS-associated strains by impairing bacterial adherence and colonization.

Finally, the hypothetical protein O3K_20405 (Hcp_VI), despite its uncharacterized function, is highly conserved among multiple EHEC strains, including O104:H4, suggesting an important role in bacterial physiology or virulence. It may be involved in processes such as adhesion, immune evasion, or toxin production [137]. Its conservation and putative involvement in critical pathogenic mechanisms make it an attractive vaccine target. Immune responses directed against Hcp_VI could impair essential bacterial functions, limiting colonization, transmission, and the risk of severe outcomes such as HUS.

Several studies have identified numerous vaccine protein candidates, particularly those focused on EHEC strains, especially *E. coli* O157:H7 [25,138,139]. However, there has been limited exploration of candidates specifically for *E. coli* O104:H4. Other studies have introduced universal proteins in pathogenic *E. coli* but at the expense of being conserved in commensal *E. coli*, thus preventing their potential as future vaccine candidates [140]. Striking the right balance, where protein candidates are conserved among pathogenic *E. coli*, but not commensal strains, is challenging. In this study, we aimed to achieve this balance by focusing on HUS-causing *E. coli* strains, both O157:H7 (EHEC) and O104:H4 (EAHEC), and ensuring divergence from different commensal strains. These four proteins fulfill this dual aim effectively.

Although O104:H4 has not caused large-scale outbreaks since 2011, its hybrid virulence profile (combining features of EAEC and STEC), multidrug resistance, and severe clinical outcomes justify its inclusion as a model strain for vaccine research. The inclusion of both O157:H7 and O104:H4 allows us to examine conserved features across distinct pathogenic lineages, potentially informing broader vaccine strategies.

Notably, past EHEC vaccine efforts have faced significant limitations. A human vaccine developed in Canada targeting O157:H7 was discontinued due to severe local reactions, including injection site abscesses. It was later reformulated for cattle use but demonstrated limited effectiveness in preventing colonization or shedding [141]. Additionally, no human EHEC vaccine has received regulatory approval to date. These challenges underscore the importance of identifying safer, broadly effective targets that can be applied across different EHEC serotypes.

The diversity of EHEC serotypes remains a critical obstacle in vaccine development. Future strategies should consider multivalent or cross-protective designs to provide coverage against a broader range of clinically relevant serotypes. Our selection of targets contributes to this direction by prioritizing antigens with high pathogenic specificity and minimal cross-reactivity with commensal flora.

Despite Stx being the primary cause of HUS and neutralizing Shiga toxin-specific antibodies potentially serving as therapeutic agents [49,50], we did not include them in our vaccine proposal due to the existence of multiple Stx types [142]. Obtaining a universally conserved sequence for Stx is challenging given the diversity of Stx types and their structural variations. Instead, our focus is on identifying stable antigens that can serve as foundational elements for either a single vaccine or a multi-vaccine candidate. The severe effects observed are largely attributed to the super-adhesion properties that enable prolonged bacterial persistence and increased secretion of Stx toxins, exacerbating the condition. Targeting this adhesion mechanism from the onset represents a strategic intervention to mitigate the downstream effects of Stx. Thus, the core aim of our vaccine strategy is to interfere with the colonization process, thereby indirectly attenuating Stx-mediated pathology.

We identified and analyzed key epitopes from selected vaccine candidates to construct a multi-epitope vaccine (MEV) designed to enhance immune activation. Rather than using whole proteins, we focused on assembling the most significant epitopes into a single construct, incorporating an internal adjuvant to further stimulate immune responses. We screened these epitopes based on their strong predicted immunogenicity to ensure effective incorporation in the MEV design. Special emphasis was placed on B-cell epitopes, reflecting the critical role of humoral immunity—particularly IgA—in combating *E. coli* infections. Of particular importance was the identification of subtype A epitopes within linear B-cell epitopes, which play a vital role in preventing bacterial adhesion to mucosal surfaces and thus blocking the initial steps of infection.

In selecting the HTL, the design incorporated predictions for IL-4 and IL-10 inducers, recognizing their fundamental roles in immune modulation. IL-4 is essential for promoting Th2 responses, which are crucial for defense against extracellular pathogens and enhancing antibody production, thereby reinforcing humoral immunity. Conversely, IL-10 serves as a potent anti-inflammatory cytokine critical for regulating immune responses and preventing excessive inflammation and tissue damage. By enhancing IL-4 and IL-10 production, vaccine constructs aim to achieve a balanced Th1/Th2 immune profile, fostering robust cellular and humoral immune responses while minimizing immunopathology. This integrated approach is supported by research indicating improved vaccine efficacy and safety outcomes through targeted cytokine modulation [143,144]. However, despite the inclusion of predictions and incorporation of PADRE sequences, which are Th inducers, in the *Ecoepv3* vaccine construct, Th1 responses predominated over Th2 responses in immune response simulations. The observed Th1 bias may be attributed to all the HTL epitopes being predicted to induce IFN-γ production, which is necessary for promoting cellular immunity, supporting Th1 polarization, and enhancing vaccine efficacy and immunogenicity [145].

We utilized four multi-epitope-based vaccine constructs, each incorporating diverse adjuvants to optimize immune response variability for each candidate. While all candidates exhibited favorable physicochemical characteristics, indicating potential for vaccine development, *Ecopev3* was selected for further advancement due to its better structure, as confirmed by Ramachandran plot analysis after refinement with GalaxyWEB. Importantly, docking studies confirmed the robust ability of *Ecopev3* to bind and activate TLR4 effectively, highlighting its ability to initiate a potent immune response.

Through analysis using various immunoinformatic tools, *Ecoepv3* has shown considerable promise as a vaccine candidate. Its robust potential for future development is underscored by its favorable characteristics identified through these assessments. However, it is important to note that this study currently lacks in vivo data in animal models, which is highly recommended for confirming the predicted efficacy observed in silico. Rigorous in vivo studies will be essential to validate the ability of *Ecoepv3* to induce immune responses effectively. Moreover, these studies will provide critical insights into further structural optimization, ensuring its suitability for advancing into preclinical and clinical trials.

## 5. Conclusions and Limitations

In this study, we introduced novel protein candidates with potential for vaccine development, sourced from the *E. coli* O104:H4 proteome and conserved within various strains of the O157:H7 serotype, ensuring coverage of some HUS-causing strains. Importantly, these proteins were selected to avoid conservation in commensal strains, thereby preserving the normal flora. They were utilized to design a multiepitope vaccine construct incorporating LBL, CTL, and HTL. This approach suggests that these candidates hold promise for enhancing immune responses.

However, this work is primarily computational and predictive in nature. The absence of experimental validation, particularly in vivo studies, limits the immediate applicability of the findings. Further functional assays and animal model testing are essential to evaluate the safety, immunogenicity, and efficacy of the proposed vaccine candidates.

## Figures and Tables

**Figure 1 diseases-13-00259-f001:**
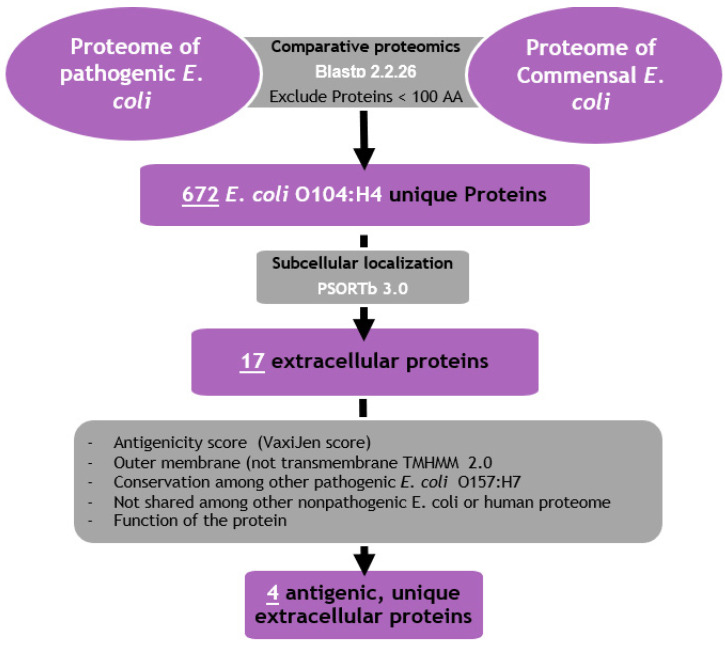
Schematic diagram illustrating the workflow for selecting target virulent *E. coli* O104:H4 proteins used in the multi-epitope vaccine in this study.

**Figure 2 diseases-13-00259-f002:**
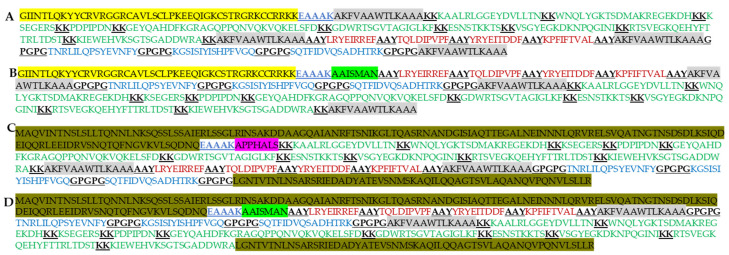
Amino acid sequences of the vaccine constructs. (**A**) *Ecoepvc1*, 399 amino acids; (**B**) *Ecoepvc2*, 393 amino acids; (**C**) *Ecoepvc3*, 534 amino acids; (**D**) *Ecoepvc4*, 532 amino acids. Human βdefensin (yellow highlight), PADRE sequence (grey highlight), linkers (black, bold, underlined), CTXB (green highlight), *S. dublin* flagellin (dark yellow highlight), RS09 (pink highlight). LBL epitopes (green font), CTL epitopes (red font), HTL epitopes (blue font).

**Figure 3 diseases-13-00259-f003:**
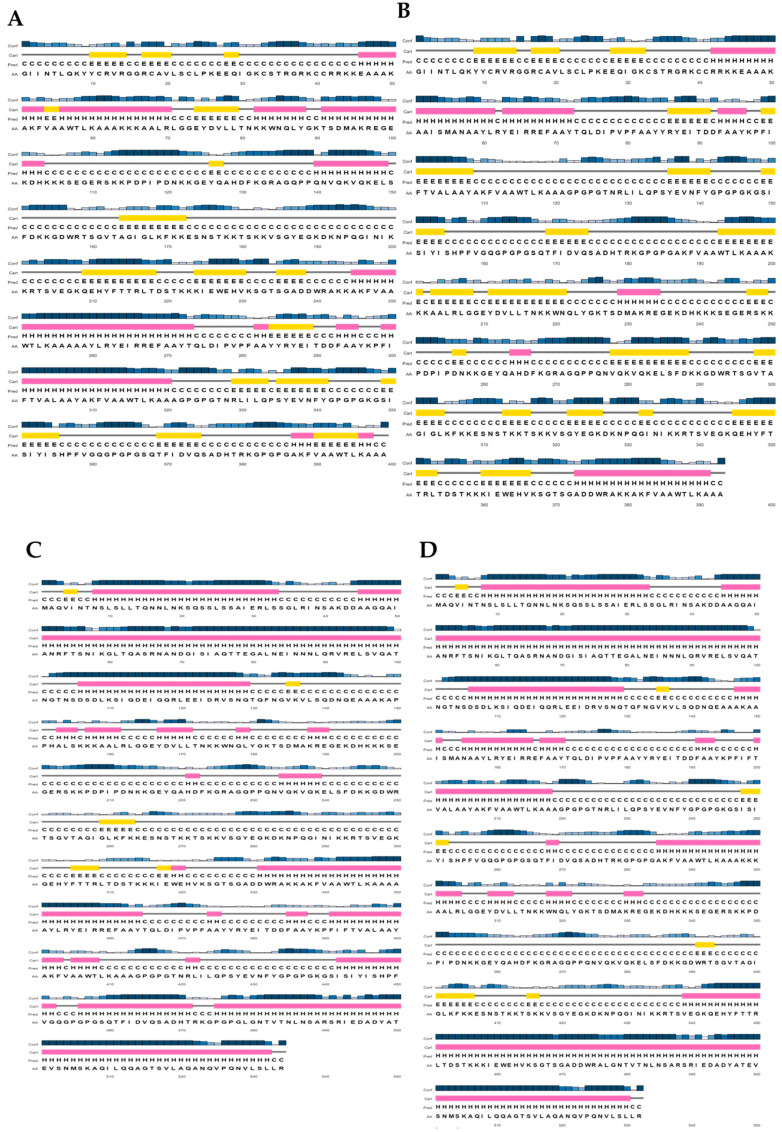
Secondary structure predictions for all vaccine constructs. PSIPRED was utilized to predict the secondary structures of MEVs: (**A**) *Ecoepvc1*, (**B**) *Ecoepvc2*, (**C**) *Ecoepvc3*, (**D**) *Ecoepvc4*. Structures are represented as strand (yellow highlight), helix (pink highlight), and coil (grey line).

**Figure 4 diseases-13-00259-f004:**
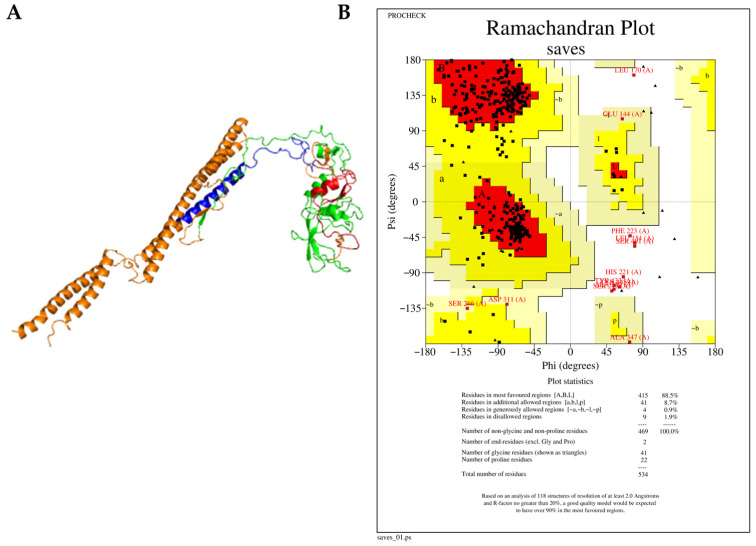
Refined tertiary structure of *Ecoepvc3*. (**A**) Predicted 3D structure of *Ecoepvc3* refined using the Galaxy Web Server. (**B**) Ramachandran plot of *Ecoepvc3* generated by PROCHECK, validating the stereochemical quality of the model.

**Figure 5 diseases-13-00259-f005:**
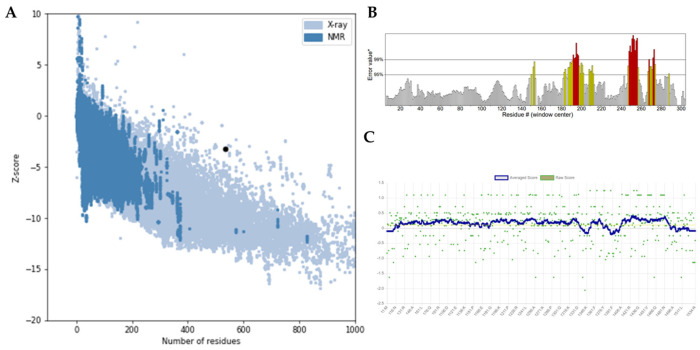
Quality analysis and structure validation of *Ecoepvc3*. (**A**) *Ecoepvc3* exhibited a Z-score of −3.21 using the Pro-SA tool. (**B**) The overall quality of *Ecoepvc3* was assessed as 80% using ERRAT software. (**C**) Verify3D analysis confirmed good compatibility between the atomic (3D) model and its corresponding amino acid sequence.

**Figure 6 diseases-13-00259-f006:**
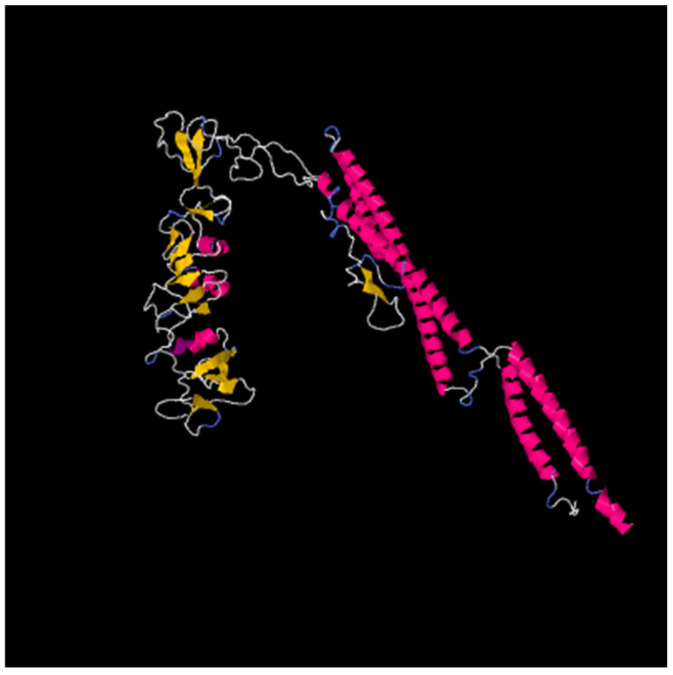
Disulfide engineering of *Ecoepvc3* construct. Predicted disulfide bonds are indicated by yellow bars.

**Figure 7 diseases-13-00259-f007:**
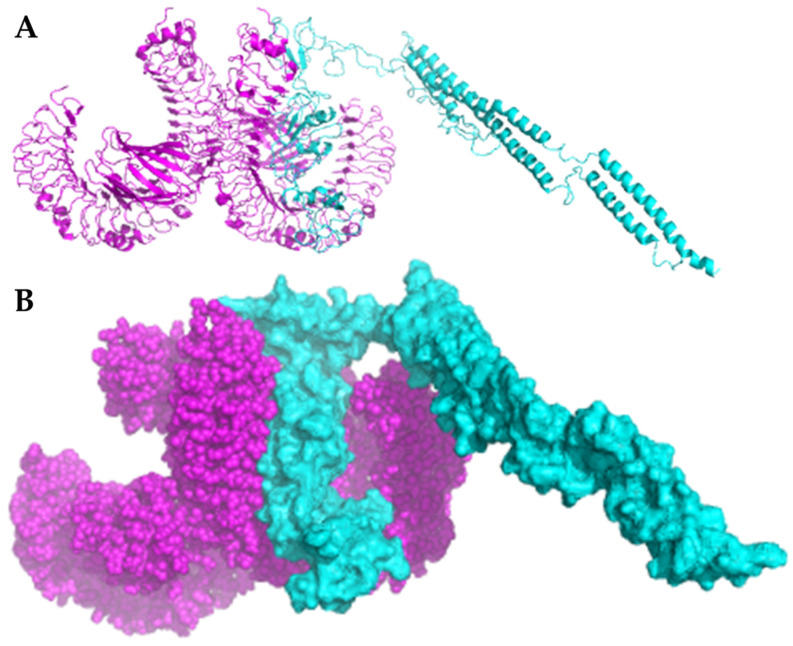
Docking results between *Ecoepvc3* and TLR4. Cyan represents the vaccine construct *Ecoepvc3*, while magenta represents TLR4. Panel (**A**) shows the ribbon view, and panel (**B**) shows the surface view.

**Figure 8 diseases-13-00259-f008:**
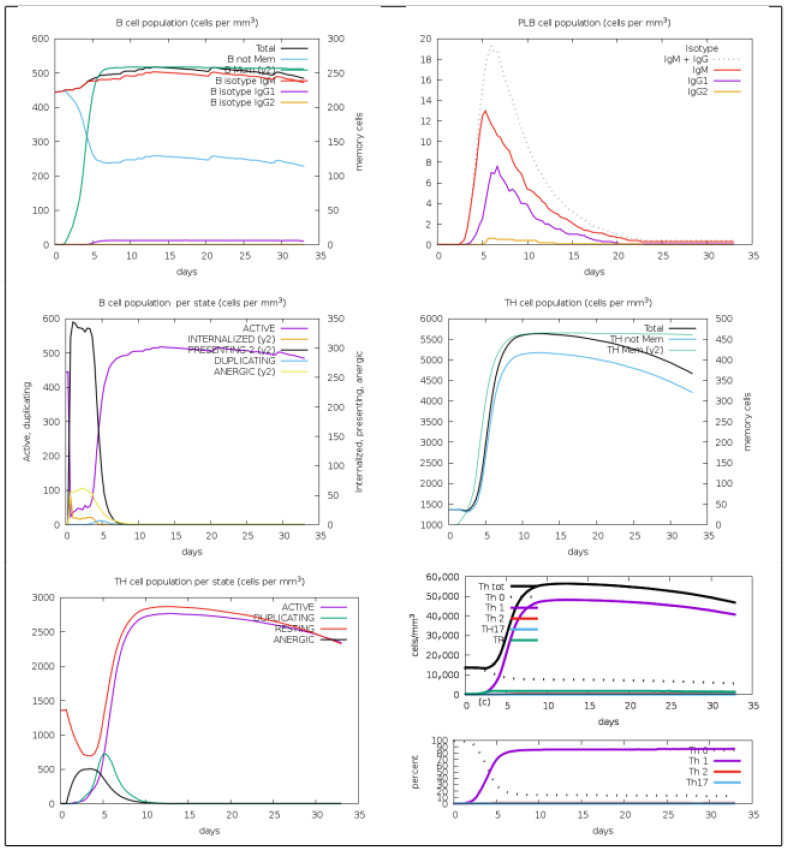
In silico stimulation of immune response using *Ecoepvc3* construct as antigen using C-IMMSUM.

**Table 1 diseases-13-00259-t001:** List of the extracellular and outer membrane proteins of *E. coli* O104:H4 strain that are not shared with commensal *E. coli* strains.

Seq ID	Name of Protein	No. of Amino Acids	Membrane Localization		Transmembrane Prediction	Signal Peptide Prediction		Antigenicity	Conservation Among Pathogenic Strains		Similarity to Commensal Strains		
			Localization	Score (Psort)	The Topology Predicted by N-Best	SignalP	LipoP	VaxiJen Score	No. of O157:H7	No. of O104:H4	K-12 MG1655	HS	W3110
407479814	long polar fimbrial protein (LpfD) [*Escherichia coli* O104:H4 str. 2011C-3493]	351	Extracellular	10	Topology = o	SP(Sec/SPI)	SPI	0.42	1	7	No similarity		
407479898	copper resistance protein B (copB) [*Escherichia coli* O104:H4 str. 2011C-3493]	299	Outer Membrane	9.93	Topology = o	SP(Sec/SPI)	SPI	0.653	1	48	No similarity		
407480277	hypothetical protein O3K_03480 [*Escherichia coli* O104:H4 str. 2011C-3493]	900	Outer Membrane	10	Topology = o	SP(Sec/SPI)	SPI	0.6301	15	13	No similarity	100%	No similarity
407480278	hypothetical protein O3K_03485 [*Escherichia coli* O104:H4 str. 2011C-3493]	362	Extracellular	9.72	Topology = o	SP(Sec/SPI)	SPI	0.6471	7	8	No similarity	78%	No similarity
407480477	serine protease pic precursor (ShMu) [*Escherichia coli* O104:H4 str. 2011C-3493]	1372	Extracellular	9.96	Topology = i34–53o	SP(Sec/SPI)	CYT	0.6182	1	9	<20%		
407480816	putative alpha-amylase, partial [*Escherichia coli* O104:H4 str. 2011C-3493]	431	Extracellular	9.45	Topology = o	OTHER	CYT	0.4347	>50	39	<50%	97%	<50%
407481100	APSE-2 prophage, transfer protein gp20 [*Escherichia coli* O104:H4 str. 2011C-3493]	488	Extracellular	9.64	Topology = o	OTHER	CYT	0.5866	None	9	No similarity	<50%	No similarity
407481484	yersiniabactin/pesticin outer membrane receptor (IRPC) [*Escherichia coli* O104:H4 str. 2011C-3493]	673	Outer Membrane	10	Topology = o	SP(Sec/SPI)	SPI	0.6608	11	8	<20%	<20%	<20%
407482061	outer membrane precursor Lom [*Escherichia coli* O104:H4 str. 2011C-3493]	241	Outer Membrane	9.93	Topology = o	SP(Sec/SPI)	SPI	0.7296	2	8	<20%	<20%	<20%
407482127	lipoprotein [*Escherichia coli* O104:H4 str. 2011C-3493]	1325	Outer Membrane	9.99	Topology = o	LIPO(Sec/SPII)	CYT	0.6884	33	46	99%	97%	97%
407482355	host specificity protein J [*Escherichia coli* O104:H4 str. 2011C-3493]	1159	Extracellular	9.64	Topology = o	OTHER	CYT	0.6137	>50	>50	83%	75%	83%
407482363	tail protein [*Escherichia coli* O104:H4 str. 2011C-3493]	220	Extracellular	9.64	Topology = o	OTHER	CYT	0.6988	None	1	78%	No similarity	78%
407482726	putative outer membrane protein Lom [*Escherichia coli* O104:H4 str. 2011C-3493]	244	Outer Membrane	8.86	Topology = o	SP(Sec/SPI)	SPI	0.8011	11	9	<20%	<20%	<20%
407483030	host specificity protein J of prophage [*Escherichia coli* O104:H4 str. 2011C-3493]	1165	Extracellular	9.64	Topology = o	OTHER	CYT	0.6174	>50	>50	82%	60%	82%
407483596	hypothetical protein O3K_20405 [*Escherichia coli* O104:H4 str. 2011C-3493]	172	Extracellular	9.71	Topology = o	OTHER	CYT	0.6139	10	7	<20%	100	<20%
407484105	serine protease pet precursor (Plasmid-encoded toxin pet) [*Escherichia coli* O104:H4 str. 2011C-3493]	1285	Outer Membrane	10	Topology = i35–57o	OTHER	CYT	0.6563	2	5	<20%	<20%	<20%
407484114	ferric aerobactin receptor [*Escherichia coli* O104:H4 str. 2011C-3493]	731	Outer Membrane	10	Topology = o	SP(Sec/SPI)	SPI	0.6267	3	36	<20%	<20%	<20%

Green highlights indicate selected proteins. Transmembrane predictions are marked as “o” (outside) and “i” (inside). Based on SignalP analysis: SP(Sec/SPI) denotes classical secretory proteins; LIPO(Sec/SPII) are lipoproteins cleaved by Signal Peptidase II; OTHER lacks classical signal peptides. LipoP analysis: SPI indicates Sec pathway secretion via Signal Peptidase I; CYT denotes non-secreted cytoplasmic proteins.

**Table 2 diseases-13-00259-t002:** List of the selected Linear B-lymphocyte (LBL) epitopes.

Name of Protein	Predicted Epitope Sequence	Start	End	Length	Software			VaxiJen Score	Antigenicity	Signal Peptide	Allergenicity	Toxicity	Virulence	Ig Subtype/Score
					ABCpred	BeriPred	BCEPRED							
Copper resistance protein B (copB)	KAALRLGGEYDVLLTN	202	217	16	√	-	-	0.7658	Antigenic	Not	Non-Allergen	Non-Toxin	Virulent	IgG/0.805
	WNQLYGKTSDMAKREGEKDH	268	287	20	√	√	√	1.272	Antigenic	Not	Non-Allergen	Non-Toxin	Virulent	-
	KSEGERS	131	137	7	Partial	Partial	√	2.4633	Antigenic	Not	Non-Allergen	Non-Toxin	Virulent	-
Long polar fimbrial protein (LpfD)	PDPIPDN	77	83	7	√	√	√	0.624	Antigenic	Not	Non-Allergen	Non-Toxin	Virulent	-
	GEYQAHDFKGRAGQPPQNVQKVQKELSFD	222	250	29	Partial	Partial	√	0.6475	Antigenic	Not	Non-Allergen	Non-Toxin	Virulent	IgG/0.854
Putative outer membrane protein Lom (LomP)	GDWRTSGVTAGIGLKF	229	244	16	√	Partial	-	1.4605	Antigenic	Not	Non-Allergen	Non-Toxin	Virulent	IgA/0.803
	VSGYEGKDKNPQGINI	78	93	16	√	√	√	1.4541	Antigenic	Not	Non-Allergen	Non-Toxin	Virulent	IgA/0.76
	ESNSTKKTS	194	202	9	√	√	√	2.3195	Antigenic	Not	Non-Allergen	Non-Toxin	Virulent	-
Hypothetical protein O3K_20405 (Hcp_VI)	KIEWEHVKSGTSGADDWRA	150	168	19	Partial	√	Partial	1.0865	Antigenic	Not	Non-Allergen	Non-Toxin	Virulent	-
	RTSVEGKQEHYFTTRLTDST	100	119	20	√	Partial	√	1.0263	Antigenic	Not	Non-Allergen	Non-Toxin	Virulent	-

√: predicted (partial) to be LBL; Partial: partially predicted to be LBL; -: not predicted to be LBL.

**Table 3 diseases-13-00259-t003:** List of the selected predicted Cytotoxic T-lymphocytes (CTL) epitopes.

Protein Name	Position	HLA	Peptide	Core	1-log50k (aff)	Affinity (nM)	%Rank	TAP IC50	TAP	Immunogenicity	VaxiJen Score	Virulence	Allergenicity	Toxicity
Copper resistance protein B (copB)	251	HLA-B2705 HLA-C0602 HLA-C0701 HLA-C0702 HLA-C1203	LRYEIRREF	LRYEIRREF	0.56	196.48	0.34	1.00	B27, B39	0.38	0.74	Virulent	Non-Allergen	Non-Toxin
Long polar fimbrial protein (LpfD)	120	HLA-A2402 HLA-B3901 HLA-B4001 HLA-B1501 HLA-C0401 HLA-C0702	TQLDIPVPF	TQLDIPVPF	0.301	1991.80	0.90	1.00	A24, B27, B62	0.17	1.11	Virulent	Non-Allergen	Non-Toxin
Putative outer membrane protein Lom (LomP)	94	HLA-B2705 HLA-B3901 HLA-C0303 HLA-C0501 HLA-C0602 HLA-C0701 HLA-C0702 HLA-C1203 HLA-C1402	YRYEITDDF	YRYEITDDF	0.507333333	1073.47	0.67	1.55	B27, B39	0.30	0.9072	Virulent	Non-Allergen	Non-Toxin
hypothetical protein O3K_20405 (Hcp_VI)	65	HLA-B0702 HLA-B0801 HLA-B3901 HLA-C0401 HLA-C1203 HLA-C1402	KPFIFTVAL	KPFIFTVAL	0.418666667	1560.28	0.88	1.23	B7, B8, B39	0.38	0.74	Virulent	Non-Allergen	Non-Toxin

**Table 4 diseases-13-00259-t004:** List of the selected predicted helper T-lymphocytes (HTL) epitopes.

Protein Name	Position	MHC	Peptide	Core	%Rank_EL	Affinity (nM)	%Rank_BA	VaxiJen Score	Antigenicity	Virulence	Allergenicity	Toxicity	INF-γ	IL4	IL10
Copper resistance protein B (copB)	216	DRB1_0101 DRB1_0102 DRB1_0103 DRB1_1201 DRB1_1302 DRB1_1501 DRB1_1503 DRB1_1601 DRB5_0202	TNRLILQPSYEVNFY	LILQPSYEV	1.03	75.20	0.59	0.85	Antigenic	Virulent	Non- Allergen	Non-Toxin	Positive	Non IL4 inducer	IL10 inducer
Long polar fimbrial protein (LpfD)	154	DRB1_0402 DRB1_0803 DRB1_1201 DRB1_1301 DRB1_1302 DRB1_1501 DRB1_1503 DRB1_1601 DRB5_0202	KGSISIYISHPFVGQ	ISIYISHPF	0.59	119.90	0.85	0.64	Antigenic	Virulent	Non- Allergen	Non-Toxin	Positive	IL4 inducer	Non IL10 inducer
Putative outer membrane protein Lom (LomP)	118	DRB1_0401 DRB1_0408 DRB1_1001	SQTFIDVQSADHTRK	FIDVQSADH	1.58	153.37	6.24	0.69	Antigenic	Virulent	Non- Allergen	Non-Toxin	Positive	IL4 inducer	IL10 inducer

**Table 5 diseases-13-00259-t005:** Summary of the physicochemical properties of all the vaccine construct.

Software used	Parameter	Vaccine Constructs			
		EcoEpvc1	EcoEpvc2	EcoEpvc3	EcoEpvc4
EXPASY ProtParam	Number of amino acids	399	393	534	532
	Molecular weight	44123.57	43452.74	58448.51	58177.17
	Theortical PI	9.92	9.89	9.67	9.63
	Total number of negatively charged residues	37	37	54	54
	Total number of positively charged residues	77	75	77	75
	Formula	C_1987_H_3131_N_563_O_562_S_7_	C_1949_H_3077_N_555_O_557_S_8_	C_2567_H_4101_N_749_O_806_S_3_	C_2551_H_4076_N_744_O_805_S_4_
	Total number of atoms	6250	6146	8226	8180
	Extinction coefficients	71195	65695	58330	58330
	Estimated half-life	30 h (mammalian reticulocytes, in vitro) > 20 h (yeast, in vivo). > 10 h (*Escherichia coli*, in vivo)			
	Instability index	24.66	25.8	30.61	30.04
	Aliphatic index	61.28	60.7	70.81	71.26
	Grand average of hydropathicity (GRAVY)	−0.712	−0.734	−0.72	−0.694
Novoprolab	Net Charge at pH 7	40.3	38.3	23.7	21.6
Protein-Sol	Solubility	0.563	0.541	0.578	0.56
SOLpro	Solubility	0.97404	0.953268	0.562518	0.760289
AntigenPro	Antigenicity	0.910422	0.891404	0.938256	0.93976
Vaxijen	Antigenicity	0.9853	0.9796	0.8414	0.8254
Allertop2	Allergenicity	Non-Allergen	Non-Allergen	Non-Allergen	Non-Allergen
TMHMM	Transmembrane domains	No	No	No	No
SignalP	Signal peptide	No	No	No	No
BlastP	Similarity to humans	11% +100% (b-defensin)	11% +100% (b-defensin)	No	hNo

**Table 6 diseases-13-00259-t006:** Prediction of N-linked glycosylation sites using GlycoPP v1.0.

Predication of N-Linked			
Position	Residue	Score	Prediction
6	NTN	−0.06	Non-glycosylated
8	NSL	−0.01	Non-glycosylated
16	NNL	−0.05	Non-glycosylated
17	NLN	−0.23	Non-glycosylated
19	NKS	0.40	Potential Glycosylated
39	NSA	0.04	Potential Glycosylated
52	NRF	−0.11	Non-glycosylated
57	NIK	0.14	Potential Glycosylated
67	NAN	−0.02	Non-glycosylated
69	NDG	−0.23	Non-glycosylated
83	NEI	−0.47	Non-glycosylated
86	NNN	−0.45	Non-glycosylated
87	NNL	−0.70	Non-glycosylated
88	NLQ	−0.51	Non-glycosylated
101	NGT	0.25	Potential Glycosylated
104	NSD	−0.30	Non-glycosylated
128	NQT	−0.15	Non-glycosylated
133	NGV	0.08	Potential Glycosylated
142	NQE	−0.29	Non-glycosylated
173	NKK	0.13	Potential Glycosylated
177	NQL	−0.39	Non-glycosylated
213	NKK	−0.26	Non-glycosylated
233	NVQ	−0.47	Non-glycosylated
267	NST	0.31	Potential Glycosylated
285	NPQ	−0.22	Non-glycosylated
290	NIK	−0.48	Non-glycosylated
420	NRL	−0.94	Non-glycosylated
431	NFY	−0.01	Non-glycosylated
481	NTV	−0.49	Non-glycosylated
485	NLN	−0.27	Non-glycosylated
487	NSA	0.04	Potential Glycosylated
504	NMS	0.77	Potential Glycosylated
523	NQV	−0.25	Non-glycosylated
528	NVL	−0.55	Non-glycosylated

**Table 7 diseases-13-00259-t007:** Prediction of O-linked glycosylation sites using GlycoPP v1.0.

Predication of O-Linked				Predication of O-Linked			
Position	Residue	Score	Prediction	Position	Residue	Score	Prediction
7	T	−0.47	Non-glycosylated	268	S	0.52	Potential Glycosylated
9	S	−0.50	Non-glycosylated	269	T	−0.41	Non-glycosylated
11	S	−0.35	Non-glycosylated	272	T	−0.46	Non-glycosylated
14	T	−0.38	Non-glycosylated	273	S	−0.20	Non-glycosylated
21	S	0.68	Potential Glycosylated	277	S	0.87	Potential Glycosylated
23	S	0.06	Potential Glycosylated	295	T	−0.59	Non-glycosylated
24	S	0.37	Potential Glycosylated	296	S	−0.57	Non-glycosylated
26	S	0.93	Potential Glycosylated	306	T	−1.10	Non-glycosylated
27	S	0.15	Potential Glycosylated	307	T	−0.73	Non-glycosylated
33	S	−0.02	Non-glycosylated	310	T	−1.01	Non-glycosylated
34	S	−0.58	Non-glycosylated	312	S	0.44	Potential Glycosylated
40	S	0.73	Potential Glycosylated	313	T	−0.77	Non-glycosylated
55	T	−0.68	Non-glycosylated	324	S	0.13	Potential Glycosylated
56	S	−0.03	Non-glycosylated	326	T	−0.01	Non-glycosylated
62	T	−0.41	Non-glycosylated	327	S	−0.24	Non-glycosylated
65	S	−0.28	Non-glycosylated	344	T	−0.95	Non-glycosylated
73	S	−0.20	Non-glycosylated	365	T	0.15	Potential Glycosylated
77	T	−0.20	Non-glycosylated	382	T	−0.06	Non-glycosylated
78	T	−0.46	Non-glycosylated	394	T	−0.58	Non-glycosylated
96	S	0.33	Potential Glycosylated	408	T	0.08	Potential Glycosylated
100	T	−0.60	Non-glycosylated	419	T	−0.23	Non-glycosylated
103	T	−0.70	Non-glycosylated	427	S	−0.19	Non-glycosylated
105	S	0.01	Potential Glycosylated	441	S	0.71	Potential Glycosylated
107	S	−0.32	Non-glycosylated	443	S	0.90	Potential Glycosylated
111	S	−0.48	Non-glycosylated	447	S	0.10	Potential Glycosylated
127	S	−0.68	Non-glycosylated	459	S	0.76	Potential Glycosylated
130	T	0.36	Potential Glycosylated	461	T	−0.56	Non-glycosylated
139	S	0.46	Potential Glycosylated	467	S	0.47	Potential Glycosylated
155	S	−0.01	Non-glycosylated	471	T	−0.53	Non-glycosylated
172	T	−1.48	Non-glycosylated	482	T	−1.37	Non-glycosylated
183	T	−0.30	Non-glycosylated	484	T	−0.50	Non-glycosylated
184	S	−0.13	Non-glycosylated	488	S	−0.05	Non-glycosylated
199	S	−0.37	Non-glycosylated	491	S	−0.55	Non-glycosylated
204	S	−0.41	Non-glycosylated	500	T	−0.82	Non-glycosylated
242	S	−0.54	Non-glycosylated	503	S	−0.32	Non-glycosylated
251	T	−0.86	Non-glycosylated	506	S	−0.71	Non-glycosylated
252	S	1.06	Potential Glycosylated	516	T	0.54	Potential Glycosylated
255	T	−0.05	Non-glycosylated	517	S	0.16	Potential Glycosylated
266	S	−0.32	Non-glycosylated	531	S	0.22	Potential Glycosylated

## Data Availability

All data are presented in this manuscript and provided as Appendix A.

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
