# Peer review of "In Silico Design of a Multiepitope Vaccine Against Intestinal Pathogenic Escherichia coli Based on the 2011 German O104:H4 Outbreak Strain Using Reverse Vaccinology and an Immunoinformatic Approach"

_diseases, 2025, doi:10.3390/diseases13080259_

Round 1

Reviewer 1 Report

Comments and Suggestions for Authors

Human vaccines for EHEC have been developed before but were not adopted.  A Canadian vaccine led to huge injection site abscesses so it was re-tooled for use in cattle, but did not work well in cattle, either.  Most importantly, there are many different EHECs - and since 2011 there have been no large outbreaks of O104:H4.  Not sure O104:H4 is best target and even if include O157 - still missing a number of major EHECs.  Should discuss this.

L20 Using antibiotics to treat EHEC not recommended as increased Shiga toxin production.  You mention this later.  Accordingly antimicrobial resistance is not a good reason for vaccine development.   Important to have a vaccine because treatment options after infection are so limited and have not really changed in the past 40 years.

L23 These are serogroups

L33 What is TLR4?

L44 EHEC mostly found in large intestine. 

L49 EHEC have intimin, STEC do not .  Would not advise using these terms interchangeably in scientific literature. 

L67 O157 also most common outbreak EHEC. 

L72 What does O104 have to do with outbreak in Georgia?  Please clarify. 

L82 How can 53 fatalities due to O104 occur in Germany and only 50 fatalities overall as determined by WHO?  These numbers do not add up. 

L81 STEC outbreak in 2011 not the same as the O104 outbreak in 2011?  Confusing. 

L83 These are statistics for O104? Normally for EHEC children and elderly most affected.  101 children??? This should be a percentage as well to be comparable to other stats. 

L85 Transmission was mainly through consuming contaminated food/fenugreek for O104:H4

L114 serogroups

Table 1 - What are the green highlighted rows?

Tables 1,2, 3, 4  - put them in landscape orientation so that they are easier to read. More space for columns. 

Figure 8  Need to move legends so that they are not covered by the graphs in some cases.

L505 Only looked at 2 selected HUS-causing variants - not all.  Need to discuss this. 

L544 Which other EHECs?  Useful to know.

L617 Some HUS-causing strains...

Author Response

Thank you so much for putting the effort and time to send me your valuable comments. Below we have responded to the comments and all the changes was marked in yellow highlight in the manuscript for better visualization.  

Comment 1: Human vaccines for EHEC have been developed before but were not adopted.  A Canadian vaccine led to huge injection site abscesses so it was re-tooled for use in cattle, but did not work well in cattle, either.  Most importantly, there are many different EHECs - and since 2011 there have been no large outbreaks of O104:H4.  Not sure O104:H4 is best target and even if include O157 - still missing a number of major EHECs.  Should discuss this.

Response 1: Thank you so much for brining this up. I have discuss it in the discussion section line 588-604

"

Although O104:H4 has not caused large-scale outbreaks since 2011, its hybrid virulence profile (combining features of EAEC and STEC), multidrug resistance, and severe clinical outcomes justify its inclusion as a model strain for vaccine research. The inclusion of both O157:H7 and O104:H4 allows us to examine conserved features across distinct pathogenic lineages, potentially informing broader vaccine strategies.

Notably, past EHEC vaccine efforts have faced significant limitations. A human vaccine developed in Canada targeting O157:H7 was discontinued due to severe local reactions, including injection site abscesses. It was later reformulated for cattle use but demonstrated limited effectiveness in preventing colonization or shedding [143]. Additionally, no human EHEC vaccine has received regulatory approval to date. These challenges underscore the importance of identifying safer, broadly effective targets that can be applied across different EHEC serotypes.

The diversity of EHEC serotypes remains a critical obstacle in vaccine development. Future strategies should consider multivalent or cross-protective designs to provide coverage against a broader range of clinically relevant serotypes. Our selection of targets contributes to this direction by prioritizing antigens with high pathogenic specificity and minimal cross-reactivity with commensal flora."

Comment 2: L20 Using antibiotics to treat EHEC not recommended as increased Shiga toxin production.  You mention this later.  Accordingly antimicrobial resistance is not a good reason for vaccine development.   Important to have a vaccine because treatment options after infection are so limited and have not really changed in the past 40 years. 

Response 2: Thank you so much. That is a valid point , we agree with you . it was edited and highlighted in yellow. "Since treatment options remain limited and have changed little over the past 40 years, there is an urgent need for an effective vaccine. Such a vaccine would offer major public health and economic benefits by preventing severe infections and reducing outbreak-related costs."

Comment 3: L23 These are serogroups

Response3: Thank you so much. We have corrected the serotype " E. coli O104:H4"

Comment 4: L33 What is TLR4?

Response 4: It is "Toll-Like Receptor 4" added in the text and highlighted in yellow line 36 

Comment 5: L44 EHEC mostly found in large intestine. 

Response 5:  Thank you so much. It was corrected and highlighted

Comment 6: L49 EHEC have intimin, STEC do not .  Would not advise using these terms interchangeably in scientific literature. 

Response 6: Thank you so much. We agree with you . we have corrected in the text to "It is important to note that enterohemorrhagic E. coli (EHEC) is a subset of Shiga toxin–producing E. coli (STEC), although these terms are sometimes used interchangeably. 

Comment 7: L67 O157 also most common outbreak EHEC. 

Response 7: It was corrected

Comment 8: L72 What does O104 have to do with outbreak in Georgia?  Please clarify. 

Response 8: It was clarified in the text "

The 2011 outbreak of E. coli O104:H4 in Europe displayed unusual virulence and lethality patterns [21]. Although a related E. coli O104 strain was reported in a 2009 outbreak in the Republic of Georgia, that strain exhibited a different molecular profile, lacked some key virulence factors (such as the Shiga toxin gene), and showed lower levels of antibiotic resistance. This indicates that while both strains share the O104 serogroup, they represent distinct lineages with different pathogenic potentials."

Comment 9: L82 How can 53 fatalities due to O104 occur in Germany and only 50 fatalities overall as determined by WHO?  These numbers do not add up. 

Response 9: Thank you so much. the numbers were revised and corrected. 

Comment 10: L81 STEC outbreak in 2011 not the same as the O104 outbreak in 2011?  Confusing. 

Response 10: We agree with you. it was corrected in the text "Approximately one month later, a smaller outbreak involving the same E. coli O104:H4 strain occurred in France"

Comment 11: L83 These are statistics for O104? Normally for EHEC children and elderly most affected.  101 children??? This should be a percentage as well to be comparable to other stats. 

Response 11: Thank  you so much. it was corrected in the text "Approximately 90% of HUS cases occurred in adults, with about two-thirds of those in females. Around 10% of HUS cases were reported in children."

Comment 12: L85 Transmission was mainly through consuming contaminated food/fenugreek for O104:H4

Response 12: Corrected in the text "The outbreak linked to the O104:H4 serotype was transmitted mainly through consumption of contaminated fenugreek sprouts serotype [22-25], indicating limited zoonotic potential."

Comment 13: L114 serogroups

Response 13: Corrected 

Comment 14: Table 1 - What are the green highlighted rows?

Response 14: Green highlights indicate selected proteins. highlighted in yellow below table

Comment 15: Tables 1,2, 3, 4  - put them in landscape orientation so that they are easier to read. More space for columns. 

Response 15: All tables are in landscape. The column adjusted as possible. 

Comment 16: Figure 8  Need to move legends so that they are not covered by the graphs in some cases.

Response 16: Thank you for your concern. Unfortunately, the website provides the curves in a fixed format that I am unable to modify.

Comment 17: L505 Only looked at 2 selected HUS-causing variants - not all.  Need to discuss this. 

Response 17: That was discussed in line 600-604 

"The diversity of EHEC serotypes remains a critical obstacle in vaccine development. Future strategies should consider multivalent or cross-protective designs to provide coverage against a broader range of clinically relevant serotypes. Our selection of targets contributes to this direction by prioritizing antigens with high pathogenic specificity and minimal cross-reactivity with commensal flora."

Comment 18: L544 Which other EHECs?  Useful to know.

Response 18: Added in the text (such as O157:H7, O26, O111, and O145)

Comment 19: L617 Some HUS-causing strains...

Response 19: Corrected in the text 

Reviewer 2 Report

Comments and Suggestions for Authors

This study explored the development of an immunogenic multiepitope vaccine (MEV) construct targeting E. coli O104:H4 using bioinformatics and immunoinformatics tools.

  1. Please verify the accuracy of the descriptions in the Introduction section. For example, lines 44-45 state: "E. coli was not recognized as a pathogen until 1982, when cases of bloody diarrhea emerged in the USA, attributed to an outbreak of the EHEC O157 strain." However, prior to the first E. coli O157:H7 outbreak in 1982, other pathogenic types (pathotypes) of E. coli were already recognized. Additionally, lines 62-63 mention "(e.g., the Sakai city incident in Japan in 1999)"; the correct year for the Sakai city incident in Japan is 1996.
  2. Table 1: As noted in the Methods section, "Proteins showing >80% coverage and >40% similarity to those in the K-12 strain were excluded." Could you clarify why some proteins with similarity exceeding 40% are still included in the table? Please also verify the entry labeled "MG1658" in the table.

Author Response

Comment 1: Please verify the accuracy of the descriptions in the Introduction section. For example, lines 44-45 state: "E. coli was not recognized as a pathogen until 1982, when cases of bloody diarrhea emerged in the USA, attributed to an outbreak of the EHEC O157 strain." However, prior to the first E. coli O157:H7 outbreak in 1982, other pathogenic types (pathotypes) of E. coli were already recognized.

Response 1: Thank you so much for catching this. 

We have edited in yellow highlight " While some pathogenic strains of E. coli had been recognized earlier, E. coli O157:H7 was first identified as a major cause of foodborne illness in 1982, following outbreaks of bloody diarrhea in the USA. This strain, a type of enterohemorrhagic E. coli (EHEC), has since become a significant public health concern" 

Comment 2: Additionally, lines 62-63 mention "(e.g., the Sakai city incident in Japan in 1999)"; the correct year for the Sakai city incident in Japan is 1996.

Response 2: Thanks for your comment. it was a typo and is corrected to 1996.

Comment 3: Table 1: As noted in the Methods section, "Proteins showing >80% coverage and >40% similarity to those in the K-12 strain were excluded." Could you clarify why some proteins with similarity exceeding 40% are still included in the table?

Response 3:

Thank you for highlighting this important point.

As described in the Methods section, proteins showing >80% coverage and >40% similarity to those in the E. coli K-12 MG1655 strain were excluded based on initial whole-genome BLAST analysis using the standalone BLAST tool. This step was conducted specifically against the K-12 MG1655 reference strain using command-line parameters.

However, this filtering process may have included some discrepancies due to limitations or potential errors in the standalone analysis. To ensure accuracy, we subsequently validated the shortlisted proteins using the NCBI online BLAST tool, comparing them against three additional commensal E. coli strains. In Table 1, we presented the percentage similarity values based on this online BLAST validation.

We acknowledge that slight discrepancies may have arisen due to differences in BLAST algorithms or database versions between the standalone and online tools. Nevertheless, the inclusion of the online validation was intended to strengthen confidence in the final list of candidate proteins.

Comment 4: Please also verify the entry labeled "MG1658" in the table.

Response 4: Thank you so much. We have verified in the table " K12 st. MG1658" and highlighted in yellow.